# Genital Prolapse in Pregnant Woman as a Presentation of Aggressive Angiomyxoma: Case Report and Literature Review

**DOI:** 10.3390/medicina58010107

**Published:** 2022-01-10

**Authors:** María Pilar Espejo-Reina, Miriam Prieto-Moreno, Marina De-Miguel-Blanc, Daniela María Pérez-Martínez, Jesús Salvador Jiménez-López, Susana Monís-Rodríguez

**Affiliations:** 1Department of Obstetrics and Gynaecology, Regional University Hospital, 29011 Malaga, Spain; prietomoreno94@gmail.com (M.P.-M.); marina.mb92@gmail.com (M.D.-M.-B.); jesuss.jimenez.sspa@juntadeandalucia.es (J.S.J.-L.); susanamonis@hotmail.com (S.M.-R.); 2Department of Anatomic Pathology, Regional University Hospital, 29011 Malaga, Spain; daperezmartinez@outlook.com

**Keywords:** aggressive angiomyxoma, pregnancy, genital prolapse

## Abstract

*Background*: Aggressive angiomyxoma is a rare entity within mesenchymal cell neoplasms, especially in pregnant women. Its main characteristic is the ability to infiltrate neighboring structures and to recur. *Case Presentation*: We present the case of a pregnant woman who debuted with a genital prolapse in the second trimester of pregnancy. She was diagnosed with bilateral ovarian teratomas and a pelvic mass of which the diagnosis could not be established until delivery. The route of delivery used was cesarean section since the genital prolapse behaved as a previous tumor. After the puerperium, the patient was referred for consultation to complete the study of the mass. The extension study was carried out with a negative result. The patient underwent surgery for tumor exeresis. Hormonal treatment was not administered according to the patient’s preferences. *Conclusions*: Aggressive angiomyxoma is a benign neoplasm that should be considered in the differential diagnosis of pelvic tumors in women. In pregnant women, the vaginal route of delivery is not contraindicated as long as the tumor does not obstruct the birth canal. The definitive treatment is surgery, preferably performed in a second stage after delivery.

## 1. Introduction

Aggressive angiomyxoma is a rare benign mesenchymal neoplasm characterized by its infiltrative capacity. It was first described in the literature in 1983 [1]. Its incidence in the general population is low, although it is known to be up to seven times higher in women than in men and it appears more frequently in the third and fourth decade of life [1,2].

The most frequent locations are mainly the vulva, vagina, female pelvis and perineum. It usually has a soft, rubbery consistency and can sometimes be confused with other benign tumors, such as lipomas, Bartholin’s gland cysts, etc. [3,4,5,6]. It is characterized by its ability to infiltrate surrounding tissues and by its ability to recur, even when free surgical margins are obtained in surgery [4,6]. The prevalence of aggressive angiomyxoma in pregnant women is even lower than in the general population and only 17 cases have been described in the literature [2].

The objective of this work is to describe a clinical case of aggressive angiomyxoma in a pregnant patient who came to a follow-up of the pregnancy and in whom the diagnosis of angiomyxoma was established after completing the study and excision of the mass in the postpartum period, and to carry out a bibliographic review of the cases published in the literature.

## 2. Case Presentation

The case of a 36-year-old pregnant woman is presented. She was referred to our center in week 8 of her pregnancy for follow-up in a high-risk obstetric clinic because she had had a history of an acute psychotic outbreak and was undergoing psychiatric follow-up. Initially, the patient was asymptomatic, and at this moment the patient took olanzapine 2.5 mg/24 h.

In the 12th week ultrasound, both ovaries were enlarged at the expense of cystic formations of mixed content with predominantly dense areas that presented low-level echogenicity and echorefringent areas that suggested the presence of fat and bone tissue. These tumors had measurements of 78 × 63 mm and 63 × 44 mm. The rest of the ultrasound findings were within normal limits (fetus with measurements according to amenorrhea, normal nuchal fold, normal ductus venosus, normal anterior placenta) and the combined screening for chromosomal diseases in the first trimester obtained a low-risk result. Tumor markers were requested and the result was negative. This, added to the fact that the patient was totally asymptomatic and the appearance of the images was highly suggestive of teratomas, ruled out the possibility of exploratory laparoscopy.

The ultrasound examination at week 20 found fetal findings within normality, with female fetal sex. The previously described tumors were visualized without changes in terms of measurements and ultrasound characteristics. The Doppler of the uterine arteries was pathological, so a follow-up was scheduled at week 26. A magnetic resonance imaging (MRI) of the pelvis was requested which described, in addition to the already known and stable lesions, a complex tumor located in the 7 × 3 × 3.5 cm vesico-vaginal space that suggested the presence of a pedunculated myoma from the cervix as a diagnostic option (Figure 1 and Figure 2).

At week 26, the patient returned to consultation for control of the Doppler study of the uterine arteries, which were still pathological, for which the sFlt1/PIGF ratio was requested, with a normal result. At this time, the patient reported a prolapsed mass through the vagina, of a soft and rubbery consistency, not painful. This mass was easily reductible, but prolapsed again when standing and during the Valsalva maneuver. That is why, initially, the doubt was raised that it could be a cystocele and a gynecological pessary was inserted for containment until the end of the pregnancy and treatment of the pelvic floor in a second stage if necessary.

The patient required hospital admission at week 34 to control a new psychotic outbreak that stabilized and was discharged 10 days later to continue the outpatient pregnancy control. At this moment, a course of corticosteroids was administered.

She had a marfanoid phenotype and was referred to Cardiology for evaluation 4 for this reason and due to a sibling’s history. The study of Marfan syndrome was negative, the maternal echocardiography was normal, and the connective tissue disease study was also negative.

Elective caesarean section was scheduled at week 37 due to the prolapsed vaginal mass, which was a previous tumor (Figure 3). During surgery, both adnexa were found to be enlarged (8 and 7 cm) at the expense of two tumors of mixed content with cystic areas, stone-like areas and greasy appearance inside, compatible with bilateral teratomas (Figure 4). A cystectomy of all the formations was performed, preserving healthy ovarian tissue in both adnexa. The lesions were sent for a deferred study. The definitive histopathological result was mature cystic ovarian teratomas.

A female was born, who weighed 2880 g and whose Apgar score was 9 at the first minute and 10 at 5 min. The postoperative period passed normally, and the patient was discharged 4 days after admission.

Subsequently, the patient was referred to the gynecology office after the puerperium, where a control transvaginal ultrasound was performed in which the previously described mass of heterogeneous appearance persisted in a difficult-to-delineate vesico-vaginal space. The internal genitalia were sonographically normal. A new pelvic MRI was requested, in which a complex, high-density 7 × 7 × 7.5 cm mass was observed, which did not seem to depend on any pelvic organ (Figure 5). The extension study was completed by computed tomography and was negative.

After completing the study, surgery was scheduled. Initially, a vaginal exploration was performed under anesthesia in the operating room, and, for the excision of the lesion, the abdominal approach was chosen, carefully dissecting the vesico-uterine plica on the anterior aspect of the vagina until the mass was located to perform the enucleation of the same. Subsequently, a redundant vaginal rhombus was cut and the vagina was fixed to the uterine isthmus-anterior cervical lip (Figure 6).

The anatomo-pathological study confirmed the diagnosis of ulcerated aggressive angiomyxoma that affects the deep margin opposite the vaginal mucosa (Figure 7).

Regarding the immunohistochemical study, the tumor had the following characteristics: Vimentin (+), Actin (+/− vessels), CD34 (+/− vessels), S-100 (−), Desmin (−), Estrogenic receptors (+) (Figure 8), Progesterone receptors (+) (Figure 9), Beta catenin (−), EMA (−), D2-40 (−), WT1 (+/− vessels), Ki 67 (< 1%).

## 3. Discussion

In the present work, a case of aggressive angiomyxoma is described, which constitutes a rare entity within benign mesenchymal tumors, in a pregnant patient, which makes it even more exceptional [2].

Our case also turned out to be complex and unique to our knowledge due to the type of presentation in the form of a genital prolapse and the concomitant existence of two adnexal tumors that made the interpretation of the findings difficult since the origin of said prolapsed mass could not be correctly established in the vagina, generating confusion about whether it was due to said mass, the adnexal tumors themselves or a pelvic organ prolapse [6].

The clinical presentation of aggressive angiomyxoma depends on its location and size, and it is often asymptomatic. In the current case, the patient did not present any symptoms prior to 12 weeks of pregnancy, at which time the tumor was detected for the first time. The differential diagnosis is important, and it should take into account benign tumors, such as lipomas, Bartholin’s cysts, pelvic organ prolapses, etc., as well as with other tumors, such as histiocytoma [6].

This type of tumor is described as “aggressive” due to its ability to infiltrate neighboring structures and its ability to recur. Distant metastatic disease, however, is rare. There are only two cases described in the literature with pulmonary metastases and six mediastinal lymph nodes [7,8,9,10]. In the present case, no metastatic disease was detected in the extension study, but in the pathological study, involvement of the deeper surgical margin of the lesion was observed, although this condition does not seem to affect the risk of recurrence, which is high per se [11].

The most appropriate imaging tests for the study of this type of lesion are ultrasound 146 and MRI. At resonance, they present a laminated pattern in T2. In T1, alternating hyper and hypointense linear areas are seen. Rarely, cystic degeneration or intratumoral vessels are seen. In the present case, both complementary tests were performed. Computed tomography is generally used as an extension study and, in the case of our patient, it was negative [2,12].

The main characteristic of aggressive angiomyxoma in the pathological study is the presence of hypocellular tissue with myxoid stroma in which stellate or spindle cells with elongated cytoplasm with little mitosis and abundant blood vessels are distinguished; this agrees with the anatomopathological findings of the present case.

The immunohistochemical study is characterized by a positive result for vimentin, hormone receptors (estrogens and progesterone) positive in up to 80% of cases, S100 (−) (rules out neural origin), and low Ki. In our case, the sample was positive for vimentin, hormone receptors were positive, S100 was negative, and Ki 67 was less than 1% [2].

Treatment consists of surgical excision of the mass, trying to achieve free edges to avoid recurrences that occur in about 30% of cases. Patient’s follow-up is extremely important since the recurrence rate is high even in surgeries with free surgical margins. In our case, the deep margin of the lesion was affected [13].

Hormone treatment with gonadotropin-releasing hormone (GnRh) agonists can be considered as adjuvant since angiomyxoma has positive hormone receptors. However, it is not clear that long-term treatment is curative or that it prevents recurrences of the disease once it is stopped [14].

During pregnancy, there seems to be an exponential growth due to hormonal influence and, although vaginal delivery is not contraindicated, if it acts as a previous tumor, a cesarean section and surgical treatment of the tumor should be performed in a second stage due to the risk of massive bleeding that it entails if it is carried out in the same act, as it was carried out in the present case [2].

In the present case, the patient was summoned to a consultation for revision and explanation of the attitude to be followed according to the decision of the expert committee. The committee, made up of gynecologists, medical oncologists, radiotherapists, radiologists and pathologists, decided an expectant attitude and follow-up with imaging tests (MRI and thoraco-abdominal CT) and markers every 6 months. In the first radiological control, recurrence versus tumor bed with low metabolic uptake was observed. Following new presentation on committee, treatment with GnRh analogues and control at 3 months were decided. The patient was advised and informed of both the benefits and the disadvantages of adjuvant treatment and accepted adjuvant hormonal treatment.

## 4. Conclusions

Aggressive angiomyxoma is a rare entity, although it must be considered for the differential diagnosis of pelvic tumors detected during pregnancy for which we do not know its etiology.

As a general rule, the maternal–fetal results are good and, in principle, the route of choice of delivery would be vaginal unless it exerts as a previous tumor since, in this case, a cesarean section would have to be performed as in our case.

The treatment of choice is surgery and close follow-up after surgery is important in order to identify possible recurrences although the most important thing of all is to assess the patient with her peculiarities and carry out an individualized treatment.

## Figures and Tables

**Figure 1 medicina-58-00107-f001:**
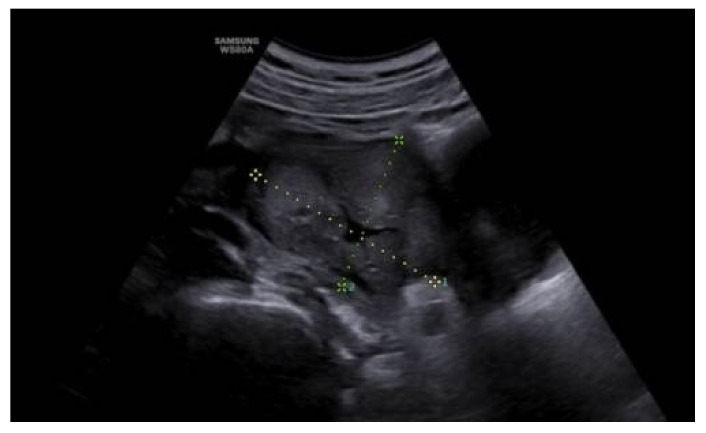
Ultrasound image of the mass at the prenatal ultrasound at 20 weeks.

**Figure 2 medicina-58-00107-f002:**
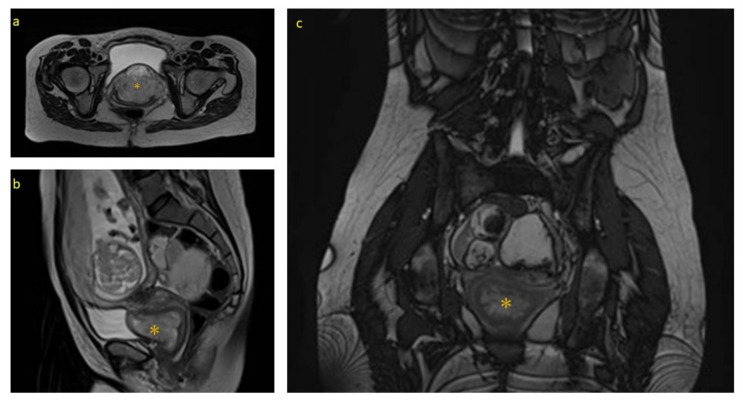
MRI (Magnetic Resonance Imaging) images during pregnancy: * Angiomyxoma (**a**) Sagittal section of the MRI; (**b**) Axial section of the MRI; and (**c**) Coronal Section of the MRI.

**Figure 3 medicina-58-00107-f003:**
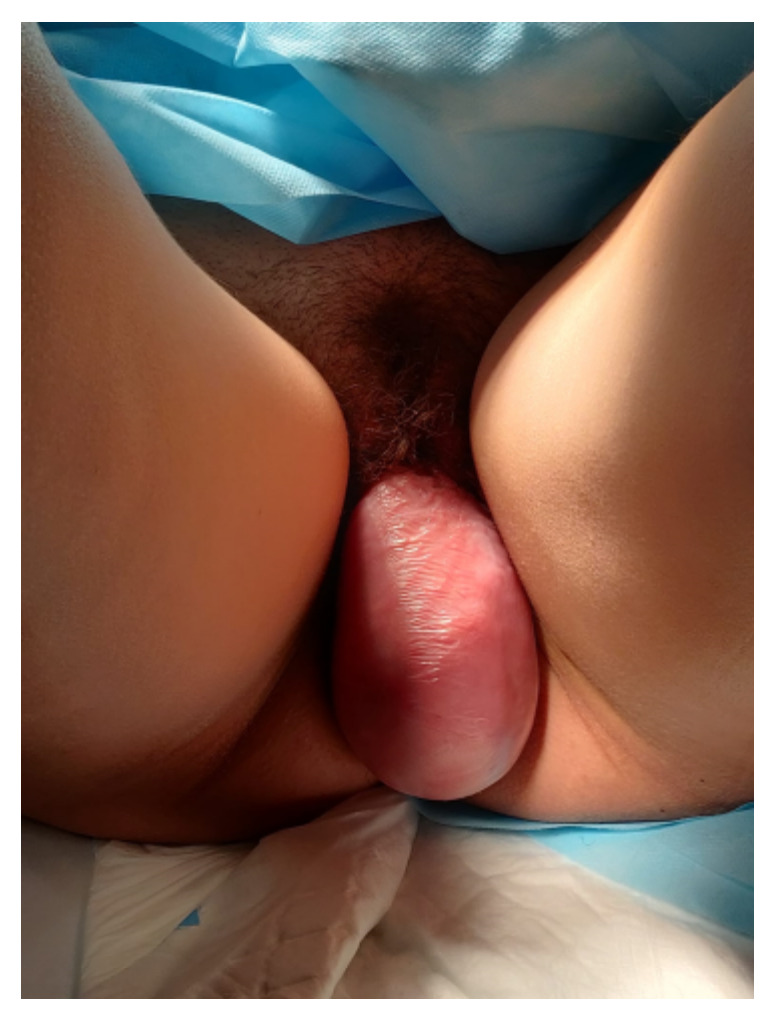
Fourth degree genital prolapse of our pregnant patient.

**Figure 4 medicina-58-00107-f004:**
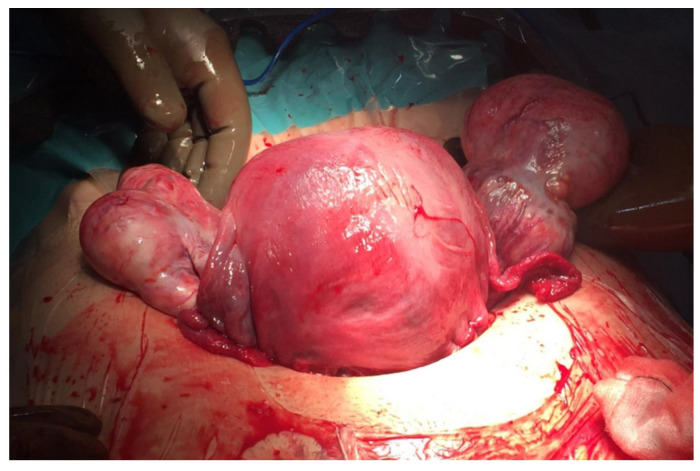
Uterus and ovaries after fetal extraction and hysterorrhaphy and before bilateral cystectomy.

**Figure 5 medicina-58-00107-f005:**
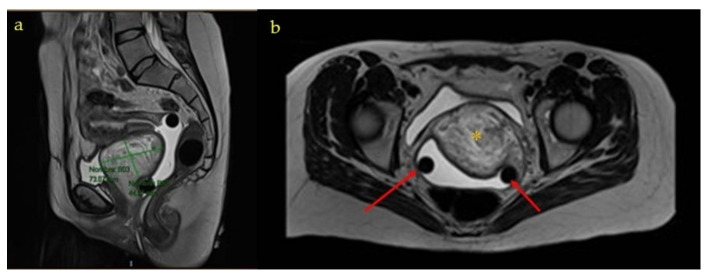
MRI images after pregnancy and before surgery: * Angiomyxoma (**a**) Axial section of the MRI. Yellow asterisk: angiomyxoma. Red arrows: gynecological pessary. (**b**) Sagital section of the MRI.

**Figure 6 medicina-58-00107-f006:**
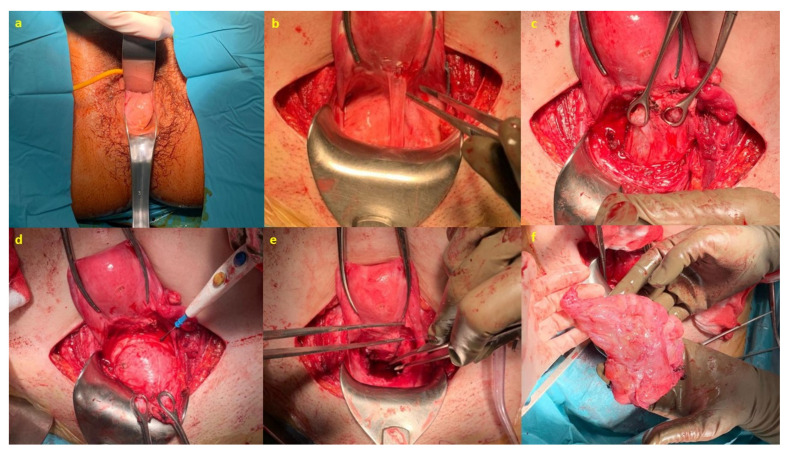
Surgery images: (**a**) Vaginal examination under anesthesia prior to the start of surgery; (**b**) Opening and dissection of the vesico-uterine plica; (**c**) Mass identification; (**d**) Enucleation of the tumor; (**e**) Clipping vagina; and (**f**) Surgical piece.

**Figure 7 medicina-58-00107-f007:**
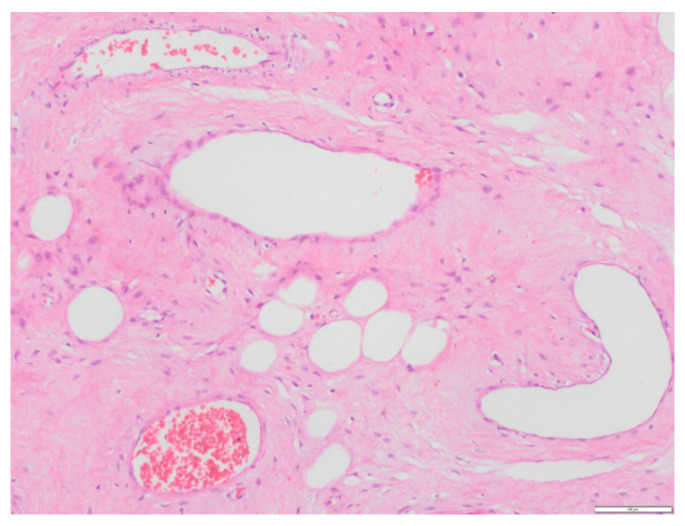
Hematoxylin–eosin stain.

**Figure 8 medicina-58-00107-f008:**
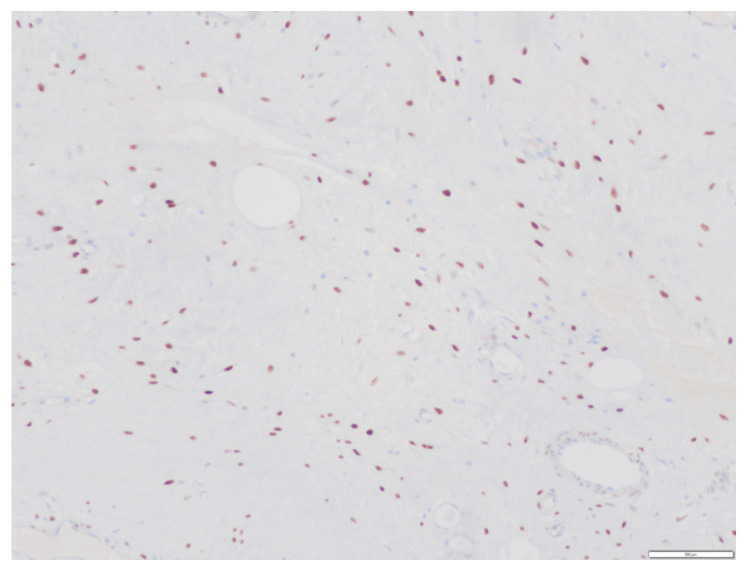
Immunohistochemistry positive estrogen receptors.

**Figure 9 medicina-58-00107-f009:**
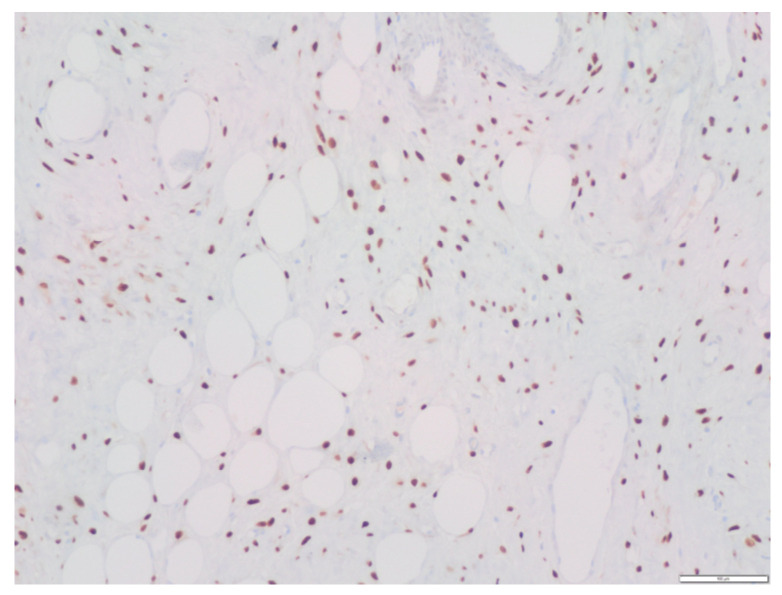
Immunohistochemistry positive progesterone receptors.

## Data Availability

Not applicable.

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
