# Peer review of "Genital Prolapse in Pregnant Woman as a Presentation of Aggressive Angiomyxoma: Case Report and Literature Review"

_medicina, 2022, doi:10.3390/medicina58010107_

Round 1
Reviewer 1 Report
Thank you for the opportunity to review María Pilar Espejo-Reina'smanuscript entitled: GENITAL PROLAPSE IN PREGNANT WOMAN AS A PRESENTATION
OF AGGRESSIVE ANGIOMYXOMA: CASE REPORT AND LITERATURE REVIEW
I have some notes on the manuscript: Line 56-64 Shouldn't the diagnostics be extended in this case?
In my opinion, the extension of diagnostics could contribute to a better
case assessment and possible earlier treatment. Was there any indication for exploratory laparoscopy in the 12th
week of pregnancy? Line 90-96 No information was given whether the intraoperative examination
was performed and what was the final histopathological result,
but only the macroscopic picture of the lesions was described
- I believe that this information should be completed. Line 117-130 What was the treatment of vaginal prolapse?
- as it is not described. After the surgery and the histopathological result,
was the oncological council convened? consisting of a gynecologist,
oncologist, clinical oncologist and radiotherapist? What were the diagnostic tests and at what time intervals
for possible dissemination? What were the operating margins? Line 174-177 Has the patient had a psychological consultation informing
her about the benefits of adjuvant treatment?
Author Response
- At week 12, when the bilateral cystic images were observed, the patient was totally asymptomatic and the appearance of such images was highly suggestive of teratomas. Thus, added to the fact that tumor markers were negative, ruled out the possibility of exploratory laparoscopy.- No intraoperative pathological examination was performed. Instead, the lesions were sent for a deferred study. The definitive histopathological result was, as we suspected, mature cystic ovarian teratomas.- The conservative treatment of prolapse was the use of different pessaries, although all were insufficient. The surgical technique consisted of laparotomical excision of the mass (Pfannestiel over a previous scar) following the plane of the same capsule, subsequently, cutting a redundant vaginal rhombus and fixing the vagina to the uterine isthmus-anterior cervical lip.
Reviewer 2 Report
The presented case of aggressive angiomyxoma in a pregnant patient is interesting.
Nevertheless, I would like to obtain some additional information, i.e.:
- Was a course of antenatal corticosteroids used for this mother? If yes, at what gestational age was it performed?
- What medications were used as “psychotropic drugs” (lines 55, 178)?
- Please describe the status of the patient’s child (especially Apgar scores, birth weight)
- What recommendations did the patient receive after gynecological operation and the pathological diagnosis of ulcerated aggressive angiomyxoma, including the date of the further gynecological visit with the ultrasound examination, recommendations for potential procreation, etc.
- Please use "Instruction for Authors" for all references.
Author Response
- A course of corticosteroids consisting of 2 doses of betamethasone 12mg / 24h, was administered in week 34 when the patient required admission to stabilize her delusional condition.- The patient took olanzapine 2.5mg / 24H throughout the gestation until week 34, when due to a new psychotic outbreak she required admission for treatment adjustment by psychiatrists, prescribing: Olanzapine 10 mg / 12h; Lorazepam 2 mg / 12h and haloperidol 5 drops / 12h until the end of pregnancy.- A female was born on 09/07/2020 at 2:30 p.m., who weighed 2880 grams and whose Apgar score was 9 at the first min and 10 at 5 min. She is currently alive and without pathology.- After gynecological surgery, the patient was summoned to a consultation for check-up, definitive anatomopathological result and explanation of the attitude to be followed according to the decision of the expert committee. The Committee, made up of gynecologists, medical oncologists, radiotherapists, radiologists, and pathologists, decided an expectant attitude and follow-up with imaging tests (MRI and thoraco-abdominal CT) and markers every 6 months. In the first radiological control, recurrence versus local tumor bed was observed, with low metabolic uptake. Following new presentation on committee, and after reviewing the literature, treatment with GnRH analogues and control at 3 months were decided. Regarding the advice for a possible procreation, at the time of the cesarean section, she was informed that a healthy ovarian remnant had been left, but given the age of the patient and the diagnosis, a new pregnancy was not advised, at least for the moment. However, the patient manifested a fulfilled birth wish.
Reviewer 3 Report
It is an important material to be published, as such a case has never been reported in the literature so far. I have a few issues to raise for the authors:
Point 1. Please insert a reference on line 37.
- Steeper TA, Rosai J: Aggressive angiomyxoma of the female pelvis and perineum. Report of nine cases of a distinctive type of gynecologic soft-tissue neoplasm. Am J Surg Pathol. 1983, 7:463-75.
Point 2. Include also the local exam of the tumor, at 26 weeks pregnancy visit. This should serve as a base for the differential diagnosis with any sort of pelvic prolapse.
Author Response
At week 26 of gestation, the patient consulted for a non painfull mass that prolapsed through the vagina. This mass was easily reducible but prolapsed again when standing and during the Valsalva maneuver.
On examination, a mass of 4-5cm, with soft and rubbery consistency, was observed protruding through the introitus, seeming to depend on the anterior aspect of the vagina. It could be easily reduced, hence the first suspected diagnosis was cystocele and an attempt to treat with a pessary was made.
The ultrasound and MRI examination did not allow the organodependence to be clearly established.
As, in addition to the tumor, the patient had teratomas, the initial diagnosis was confusing because it was not clear whether the prolapsed in the vagina was the tumor later described, the teratomas, or any other structure such as fibroids.
The patient had a marfanoid phenotype and tissue hyperlaxity was considered to be the cause of prolapse.